# Understanding the Effect of Traffic Congestion on Accidents Using Big Data

Santiago Sánchez González, Felipe Bedoya-Maya * and Agustina Calatayud

Transport Division, Inter-American Development Bank, Washington, DC 20577, USA;
santiagosa@iadb.org (S.S.G.); mcalatayud@iadb.org (A.C.)
* Correspondence: fbedoyamaya@iadb.org; Tel.: +1-202-390-9685

**Abstract:** Understanding the temporal and spatial dynamics of traffic accidents are a key determinant in their mitigation. This article leverages big data and a Poisson model with fixed effects to understand the causality of traffic congestion on road accidents in ten cities in Latin America: Bogota, Buenos Aires, Lima, Mexico City, Montevideo, Rio de Janeiro, San Salvador, Santiago, Santo Domingo, and Sao Paulo. Analyzing over 10 billion observations in 2019, results show a positive non-linear causality of congestion on the number of accidents. Overall, the results suggest that a 10% reduction in traffic delay would reduce accidents by 3.4%, equivalent to over 72 thousand traffic accidents. Sao Paulo and Mexico City would be particularly benefited, with reductions of 5.4% and 4.7%, respectively. The results of this paper aim to support policymakers in emerging economies in implementing measures to reduce congestion and, with it, the related direct and indirect costs borne by societies.

**Keywords:** traffic accidents; congestion; big data; Latin America

## 1. Introduction

Traffic congestion is ubiquitous in large cities around the world; where it leads to increased air pollution, vehicle noise, and travel time for private and public transportation [1]. These challenges reduce the wellbeing of both road users and urban populations [2]. Congestion costs in Europe were estimated at over €200 billion in 2016, which was equivalent to 1.4% of the region's GDP [3]. In the United States, such costs amounted to as much as US $160 billion in 2015 in wasted time and fuel [4] and it was expected to grow steadily to US $186.2 billion by 2030 [2]. Recent studies focusing on developing countries showed that in 2019, direct congestion costs in Sao Paulo were equivalent to the annual amount that the city invested in healthcare, and twice the annual amount that Buenos Aires and Mexico City invested in education [5]. Mitigating congestion is therefore among the main goals of transportation policy [6].

Along with direct economic costs, the time lost on a congested road that could be spent on other activities, urban congestion is also associated with increased indirect costs, including adverse health effects on individuals [7]. Wang et al. found that ten additional minutes of commuting time due to congestion is associated with a 0.8% higher chance of suffering from depression [8]. More broadly, the World Health Organization (WHO) suggests that congestion is related to higher levels of fatigue, alterations to social behavior (increased anxiety, for example), communication difficulties, and sleep disruption, ultimately impeding societies' sustainable development [9].

Among the indirect costs of congestion, extant literature has analyzed the relationship between congestion and accidents. Indeed, traffic accidents are one of the main challenges in achieving sustainable mobility [10]. Various studies have shown mixed results on this relationship. Based on a comprehensive literature review of more than 70 studies between 1937 and 2019, Retallack and Ostendorf [11] suggest four types of relationships: (i) A positive linear relationship between the volume of traffic and the total number of accidents [12]; (ii) a positive U-shaped relationship where higher levels of accidents are

found with both low and high congestion levels [13]; (iii) an inverse U-shaped relationship where the highest level of accidents is reached with an average level of congestion [14]; and (iv) a negative relationship between congestion and fatal accidents, where the effect caused by the speed reduction on accidents outweighs the effects attributed to a higher number of cars on the road [15].

Despite being a long-dated topic of interest in academic literature, there is still no agreement on the way that the amount of congestion and quantity of accidents are related. Scholars have faced two main barriers in addressing this question: (i) The relationship between congestion and quantity of accidents is influenced by a number of circumstantial factors such as weather conditions [16], urban or rural environment [17], travelers' attributes [18], and road design [19]; and (ii) the relationship is bidirectional, meaning that while it is expected that congestion influences the quantity of accidents, similarly accidents also create congestion (also known as reverse causality) [20]. The absence of temporally and spatially disaggregated data on congestion, the number of accidents, and the various factors influencing the relationship have made it difficult for previous studies to address endogeneity challenges in their methodological strategies [21]. The approaches most widely adopted use indicators to control for specific effects, including the volume-to-capacity ratio (v/c). However, the effect is still influenced by other circumstantial factors, and mixed results have been reported by the literature (see for instance [15,16,22]).

Novel technologies for traffic management such as Automated Vehicle Identification (AVI), Remote Traffic Microwave Sensors (RTMS), Bluetooth traffic sensors, and mobile navigation platforms are now generating large amounts of data to explore congestion and accident patterns at an unprecedented level of granularity [23]. Particularly, the widespread adoption of GPS-equipped mobile phones and the development of mobile applications for trip planning make them a plausible option for collecting massive amounts of traffic and other related information at a relatively lower cost, with higher update frequency, and broader coverage [24]. Despite this potential, to our knowledge, there is little research that leverages on this data to explore the relationship between urban congestion and the quantity of traffic accidents.

This study uses mobile-generated big data on urban traffic and applies a fixed-effect Poisson regression model with an instrumental variable to overcome the main barriers faced by previous research when analyzing the relationship between urban congestion and quantity of accidents. The methodological strategy applied allows for the effect that congestion has on the quantity of accidents to be isolated, controlling for fixed effects in the selected urban areas. This paper focuses on Latin America, a region often overlooked in academic literature, yet hosts four out of the ten most congested cities in the world [25]. Over 10 billion observations were collected from the mobile platform Waze during 2019 for the ten largest cities in Latin America: Bogota, Buenos Aires, Lima, Montevideo, Santo Domingo, Sao Paulo, Mexico, Rio de Janeiro, San Salvador, and Santiago. Moreover, an instrumental variable approach is followed to address the endogeneity between congestion and accidentality, therefore allowing researchers to estimate the impact of congestion on the quantity of accidents.

It is expected that the results of this paper will support policymakers in emerging economies, and especially in Latin America, in implementing measures to reduce congestion and, with it, the related direct and indirect costs borne by their respective communities. Latin America already holds 4 of the 10 most congested cities in the world (Bogota, Mexico City, Rio de Janeiro, and Sao Paulo) [26]. By 2030, motorization rates are expected to increase by almost 40% [25]. Preliminary results from studies analyzing the short-term impacts of the Covid-19 pandemic show a modal shift towards private cars [27]. In this context, bolder policies are needed to revert the projected negative trends for the region.

This paper is organized as follows: Section 2 presents the methodology for measuring traffic congestion and assessing its impact on accidentality; Section 3 presents the results, Section 4 discusses results and their implication; and finally, Section 5 summarizes the conclusions.

## 2. Materials and Methods

To measure and characterize congestion and traffic accidents in the ten selected cities, we retrieved data from Waze for the period between 1 January to 31 December 2019. Waze is a mobile navigation application with over 130 million monthly active users in 185 countries [28]. In the field of transportation research, Waze data have been used to measure the impact of sports megaevents on traffic [29], analyze road safety [30], reposition police agents [31], and estimate vehicle speed on urban corridors [32]. According to Hoseinzadeh et al. [33], Waze, as a data source, remarkably replicates actual speed and jam length. Goodall and Lee [30] evaluated the accuracy of Waze data by observing a 2.7-mile corridor on a major urban freeway in Virginia, finding that, provided that the data were filtered for spatial and temporal proximity to avoid double counting, it replicated the information collected by other means such as traffic detection cameras and policy reports.

Waze collects traffic data in two ways:

- Alerts: Each Waze user self-reports alerts noticed on the road. Once an alert is reported, other users validate it by reporting on the mobile app if the alert is still present. Based on the information received by users, Waze calculates a reliability factor between 1 and 10, 10 being the most reliable. Users can report three kinds of alerts: (i) Accident, which stands for collisions of every type; (ii) hazard, which is a type of alert that can be reported for stranded vehicles or objects on the road, adverse weather conditions, and floods, among others; and (iii) road Closed, which stands for track closures due to demonstrations, events, maintenance, and others.
- Jams: This dataset is retrieved actively by Waze through smartphones' GPS signals. When the API identifies a significant group of vehicles circulating at an irregular speed in contrast to free-flow speed, it classifies it as a jam. For each jam, Waze collects information on average speed, expected delay it would take to cross the road compared to free-flow conditions, geographical coordinates, and jam length. Information on road status is updated every 2 min.

The process we applied to build the database is illustrated in Figure 1. The geographical areas selected for data collection are shown in Figure 2. To avoid biases in our data collection, we gathered data on alerts irrespective of the existence of a jam. We selected alerts with a reliability level of 5 or higher. Given that several alerts may correspond to the same event, alerts were filtered according to the following spatial criterion: Alerts of the same type reported within a radius of 20 m in a margin of less than 20 min were considered to be the same alert. After this processing, we obtained a database of approximately 85 million unique alert records for the ten cities analyzed in this study.

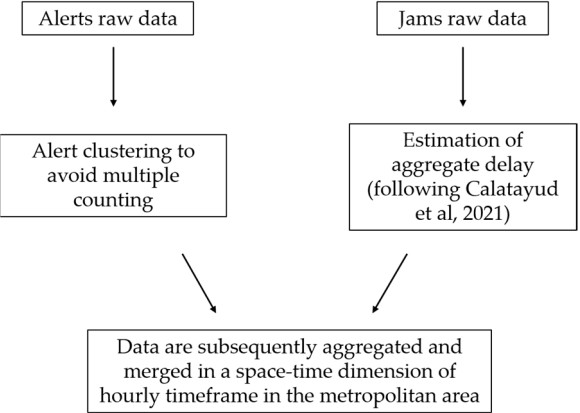

**Figure 1.** Database structure.

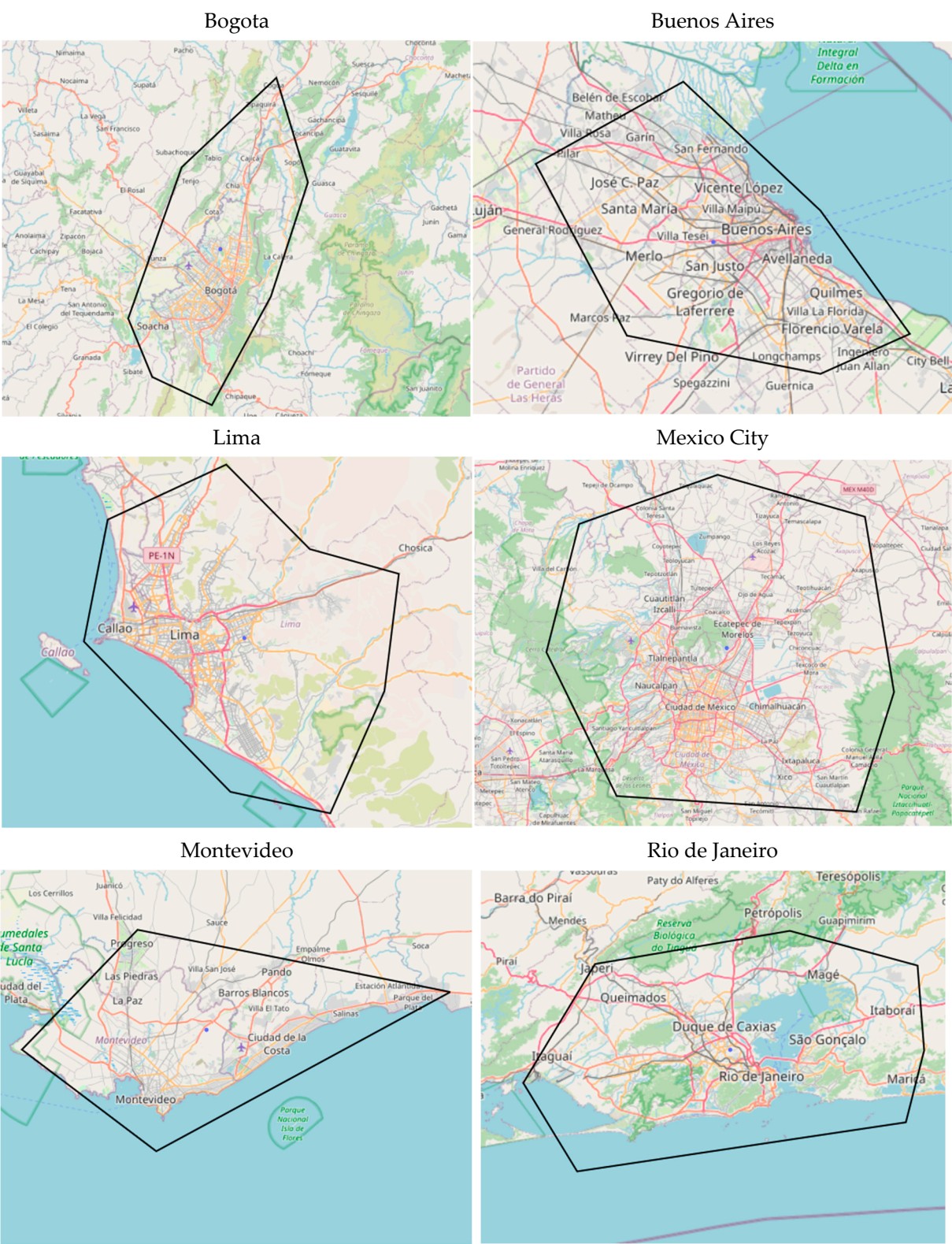

**Figure 2.** *Cont.*

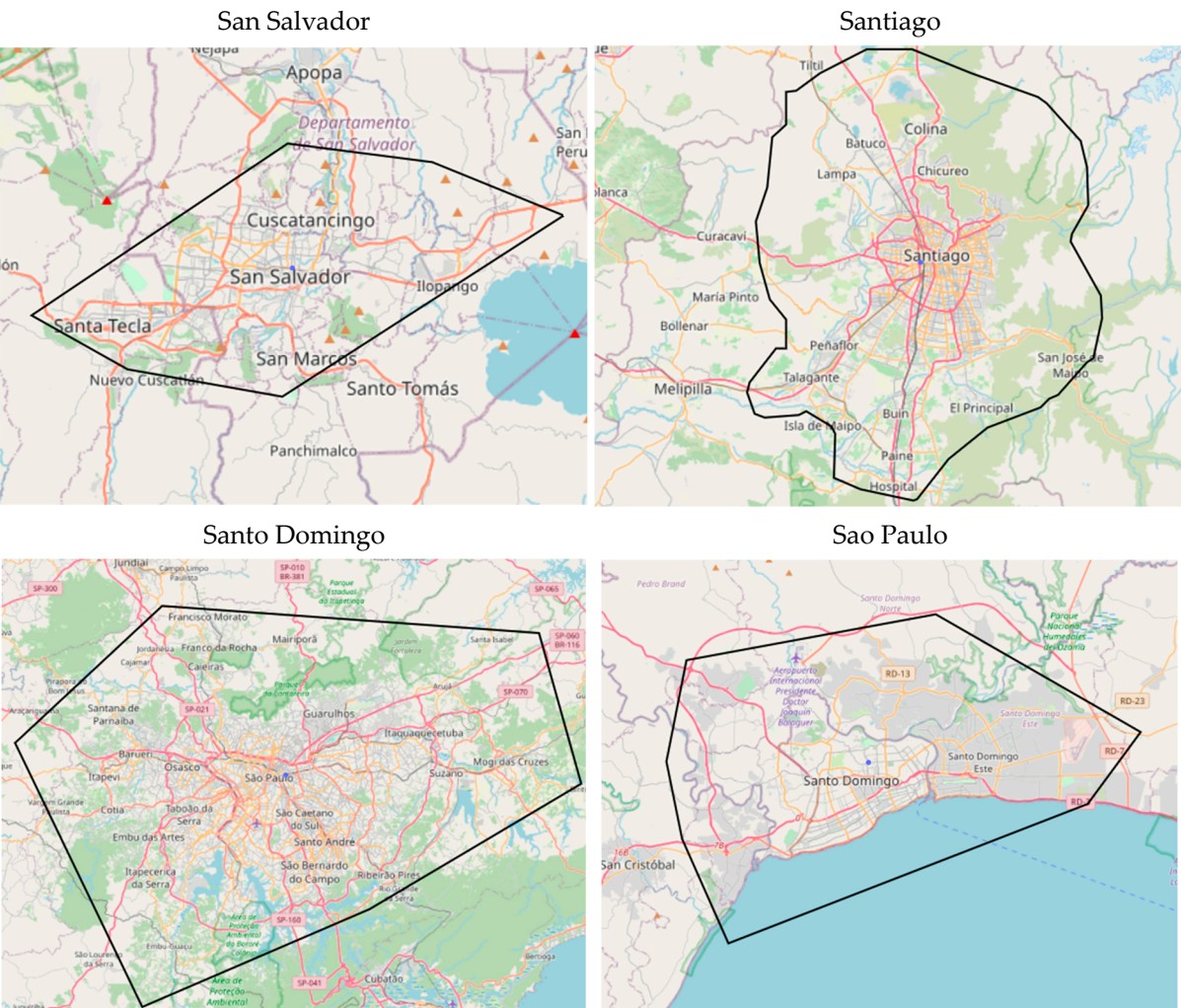

**Figure 2.** Areas under analysis (selected cities, 2019).

We collected over 10 billion observations of jams from Waze for the metropolitan areas. Following Goodwin [34], we defined congestion as the impediment that vehicles impose on each other, due to the speed–flow relationship, in conditions where the use of the transport system approaches its capacity limit. We estimated aggregate delay for the selected cities according to the methodology developed by Calatayud et al. [5], who used a two-phased methodology that included a neural network model to build the road network in each city; and a model to calculate delay based on the characteristics of Waze data.

Next, to uncover the causality of congestion on traffic accidents, the database was organized as a panel data. We aggregated data retrieved for 2019 at the city level and classified it according to an hourly time frame. By looking into 2019 data, we were able overcome the challenges posed by the atypical mobility events taking place during the COVID-19 pandemic. We estimated a Poisson model with fixed effects by city as follows Equation (1):

$$\lambda_{it} = E(Y_{it}|X_{it}) = e^{X_{it}\theta + \delta_i} \tag{1}$$

where subscripts $i$ and $t$ refer to city and date in the hourly time frame; $Y$ stands for the number of traffic accidents in city $i$ at hour $t$; $\lambda$ represent the average number of traffic accidents; $\delta$ are the city fixed effects; $X$ is a matrix that contains the independent variables of the model among which are the natural logarithm of alerts of hazards, namely floods, weather hazards, objects in the road, and stranded vehicles; Monday, Tuesday, Wednesday, Thursday, Friday, Saturday, and Sunday are dummy variables that take the value of 1 in

case of being the respective day of the week and 0 elsewise; and delay, which refers to the extra time vehicles take to go through the congested road compared to the time it would take under free-flow conditions.

The Poisson distribution function proposed to execute the model was stated as follows Equation (2):

$$pr(Y_{it} = y_{it}|X_{it}) = \frac{\lambda_{it}^{y_{it}} * e^{-\lambda_{it}}}{y_{it}!} = \frac{e^{(X_{it}\theta + \delta_i)y_{it}} * e^{-e^{X_{it}\theta + \delta_i}}}{y_{it}!} = \frac{e^{X_{it}\theta y_{it}} * e^{\delta_i y_{it} - e^{X_{it}\theta} * e^{\delta_i}}}{y_{it}!} \quad (2)$$

From Equation (2), the joint distribution function can be obtained as Equations (3) and (4)

$$pr(Y_{i1} = y_{i1}, \ldots, Y_{iT_i} = y_{iT_i}|X_{it}) = \prod_{t=1}^{T_i} \frac{e^{X_{it}\theta y_{it}} * e^{\delta_i y_{it} - e^{X_{it}\theta} * e^{\delta_i}}}{y_{it}!} \quad (3)$$

$$pr(Y_{i1} = y_{i1}, \ldots, Y_{iT_i} = y_{iT_i}|X_{it}) = \left(\prod_{t=1}^{T_i} \frac{e^{X_{it}\theta y_{it}}}{y_{it}!}\right) e^{[-e^{\delta_i} \sum_t e^{X_{it}\theta} + \delta_i \sum_t y_{it}]} \quad (4)$$

where based on the methodology proposed by Wooldridge [35], the likelihood function to be maximized is Equation (5):

$$L = ln \prod_{i=1}^{T_i} \left[ \frac{(\sum_t y_{it})!}{\prod_t y_{it}!} \prod_{t=1}^{T_i} p_{it}^{y_{it}} \right] \quad (5)$$

where $p_{it}$ is defined as in Equation (6):

$$p_{it} = \frac{e^{X_{it}\theta}}{\sum_t e^{X_{it}\theta}} \quad (6)$$

Equation (8) is the model to be estimated as to identify the effect of congestion on the quantity of traffic accidents. As mentioned in the Section 1, one of the main barriers faced by previous research was the endogeneity problem due to reverse causality between congestion and traffic accidents. To overcome this challenge, the identification strategy proposed takes advantage of time occurrence and the non-possible scenario in which current accidents could have affected congestion level in the past, as it is commonly used in time series analysis. We follow Reed [36], who concluded that when dealing with reserve causality it is best to use lagged variables as instruments if they satisfy both the exclusion restriction and are strong enough instruments. In this research, recurrent congestion supports the strength of the instrument, given that congestion tends to show similar patterns considering the respective days and times of the week. Moreover, external factors that could impact accident quantity such as road network capacity, travelers' characteristics, and level of urbanity do not vary in one year and are therefore controlled by the fixed effects used in the modeling. From this, the current delay is instrumented by its value lagged for a week, i.e., the delay that took place at the same day and hour one week before, as shown in Equation (7). Additionally, we control by the instrument roads closed (Road Closed), and fixed effects by hour and week:

$$\text{delay}_{it} = \beta_o + \alpha_i + \omega_t + h_t + \beta_1 \ln(\text{accident}_{it}) + \beta_2 \text{delay}_{it-164} + \beta_3 \ln(\text{RoadClosed}_{it-1}) + \beta_4 \ln(\text{hazard}_{it-1}) + \mu_{1it} \quad (7)$$

$$\ln(\text{accident}_{it}) = \gamma_o + \delta_i + \gamma_1 \text{delay}_{it} + \gamma_3 \ln(\text{hazard}_{it-1}) + \sum_{\text{dow}=1}^{7} \gamma_{\text{dow}+3} \text{dow}_t + \mu_{2it} \quad (8)$$

Equations (7) and (8) represent, respectively, the first and second stages of the two-stage instrumental variable estimation process. The process entails using the following variables: $\ln(\text{RoadClosed})$; $\ln(\text{hazard})$; $h$, which stands for a dummy matrix that takes the value of 1 in case of being the corresponding hour and zero otherwise; and, more

importantly, delay lagged to the same hour and day of the previous week. Furthermore, $\alpha$ and $\delta$ stand for a city's fixed effects; $\omega$ includes time series fixed effects over a weekly basis; *dow* represent all the dummies for each of the days of the week; and $\mu$ stands for the error term for both equations. Finally, both ln(RoadClosed) and ln(hazard) are introduced in the model lagged in time.

### 3. Results

Almost two million traffic accidents were reported in 2019 in the ten cities under analysis (Table 1). Seventy percent (70%) of the total accidents were concentrated in three cities, Mexico City, Bogota, and Sao Paulo. In contrast, Montevideo and Santo Domingo accounted for less than 3%. On average, there were almost 20 traffic accidents per thousand residents. San Salvador was the city with the highest number of accidents: 51 accidents per thousand residents. Santo Domingo and Buenos Aires were the cities that registered the lowest number of accidents in 2019, with 6.6 and 5.7 accidents per thousand residents, respectively. Overall, more than 3 billion hours were lost in congestion. Sao Paulo and Mexico City accounted for the highest delays, with more than 702 and 647 million hours lost, respectively. The table also shows the number of hazards and roads closed reported in each city, with hazards being on average ten times higher than roads closures.

**Table 1.** General statistics (selected cities, 2019).

| City | Accidents (Thousands) | Accidents per Thousand Habitants | Delay (Million Hours) | Hazards (Thousands) | Road Closed (Thousand) |
|---|---|---|---|---|---|
| Bogota | 465 | 42.9 | 335 | 814 | 129 |
| Buenos Aires | 86 | 5.7 | 305 | 611 | 101 |
| Lima | 136 | 12.9 | 384 | 474 | 136 |
| Mexico City | 534 | 24.5 | 647 | 2377 | 191 |
| Montevideo | 29 | 16.7 | 79 | 93 | 84 |
| Rio de Janeiro | 136 | 11.0 | 312 | 661 | 25 |
| San Salvador | 57 | 50.7 | 37 | 119 | 1 |
| Santiago | 130 | 19.1 | 194 | 970 | 22 |
| Santo Domingo | 21 | 6.6 | 76 | 90 | 4 |
| Sao Paulo | 405 | 18.4 | 702 | 2641 | 89 |
| Total | 1999 | 19.0 | 3071 | 8850 | 780 |

Source: All tables and figures throughout the manuscript are own elaboration using data from Waze.

The daily average number of accidents for each city and the respective 95% confidence intervals are shown in Figure 3. In 2019, there were more than 1000 accidents per day in Mexico City, Bogota, and Sao Paulo, while the remaining cities reported less than 500 accidents.

When considering the temporal distribution of traffic accidents by day of the week and hour of the day, a common pattern can be observed for all ten cities (Figure 4). The number of accidents increased by the end of the week, especially on Thursdays and Fridays. Specifically, Friday accounted for almost 19% of the total accidents while Sundays were the day with the least number of accidents. The afternoon (between 3 pm and 8 pm) was the time during the day when most traffic accidents occurred, accounting for almost 30% of the total. Bogota, Mexico City, Santiago, and San Salvador also showed a significant number of accidents during the morning (between 7 am and 9 am).

The period between 7 am and 9 pm was considered for the analysis on the relationship between congestion and quantity of accidents, given that almost all accidents took place during this time frame. By doing so, we excluded atypical events that could influence results. In Figure 5, aggregate traffic accidents are shown on the left axis, while aggregate delay is shown on the right axis, for all weeks of 2019 and for the different cities under analysis. In general, the two variables showed similar trends, suggesting that the periods with the largest delays tended to be accompanied by an increasing number of accidents.

The overall Pearson correlation coefficient—which shows the strength of linear association between two variables—is 0.81, and it is statistically significant. When considering cities individually, the highest correlation values were observed in Bogota, San Salvador, and Santo Domingo (0.91), while the lowest correlations corresponded to Rio de Janeiro and Montevideo (0.61 and 0.67, respectively). Correlations were also positive and statistically significant at the individual level.

The spatial correlation between congestion and accidentality is shown in Figure A1 in the Appendix A. Notice that traffic accidents not only covariates in their temporal dimension, but also in their geographical dimension. As expected, the areas with the largest amount of traffic accidents are the main avenues and highways, which also show the highest delays. This correlation is relevant in the context of the model proposed, given that the data was aggregated at the city level to execute the model, with the objective of finding the overall effect of traffic congestion on the quantity of accidents.

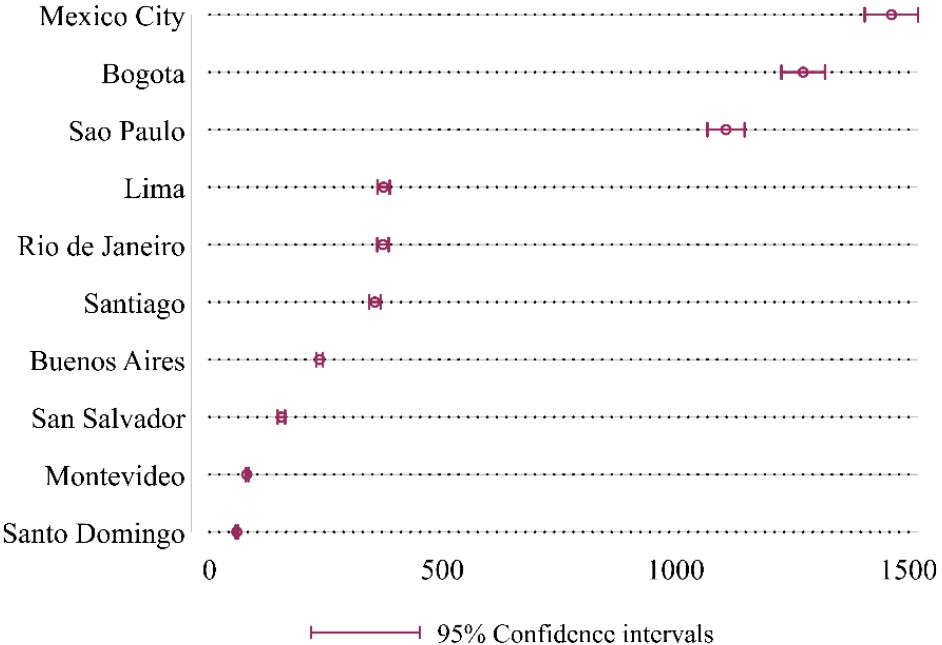

**Figure 3.** Traffic accidents daily average (selected cities, 2019).

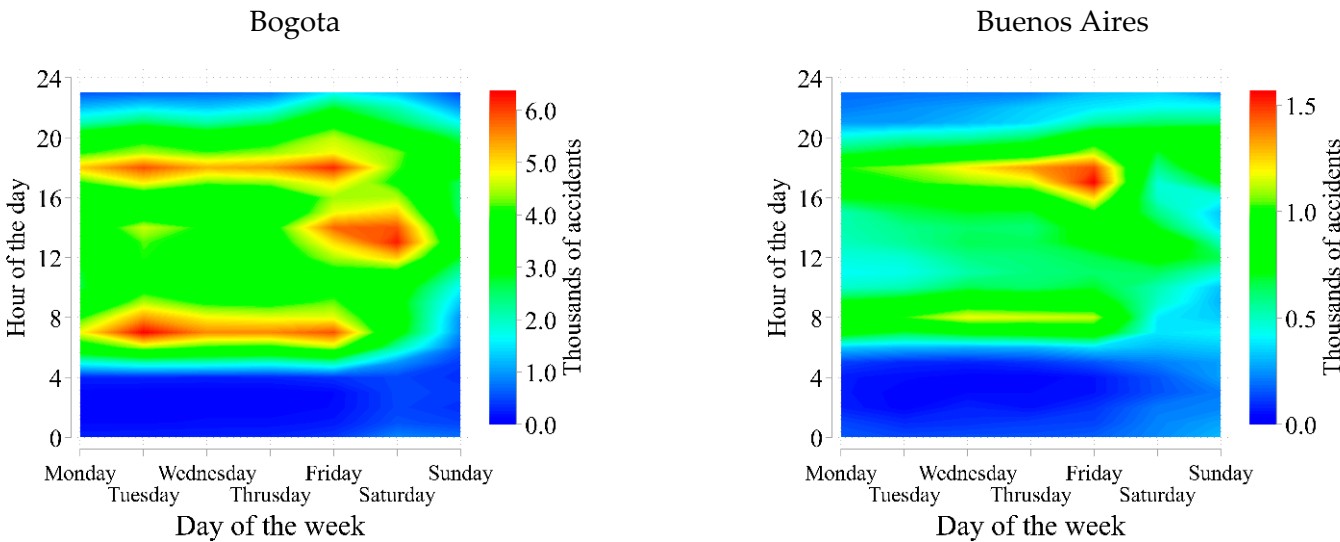

**Figure 4.** *Cont.*

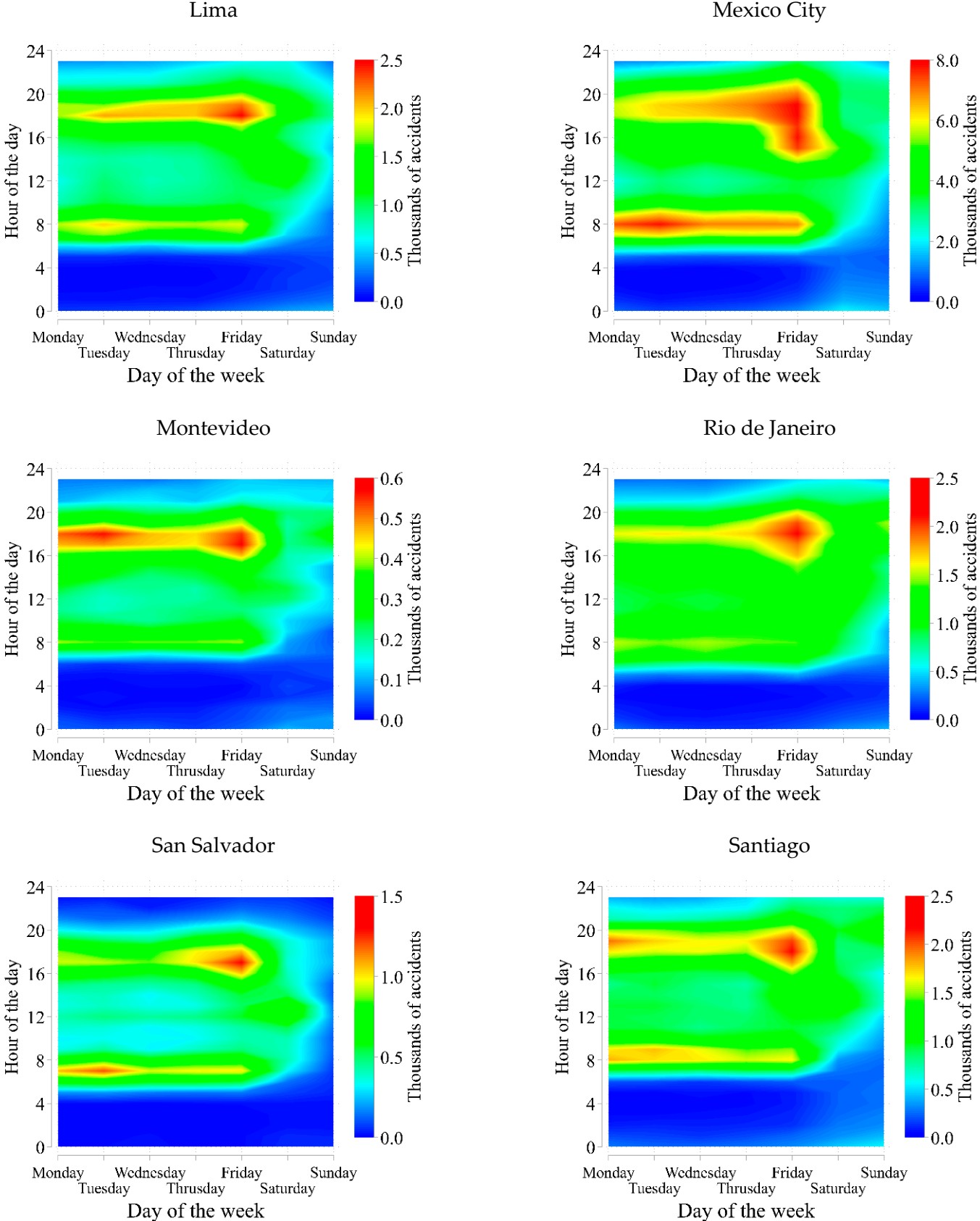

**Figure 4.** *Cont.*

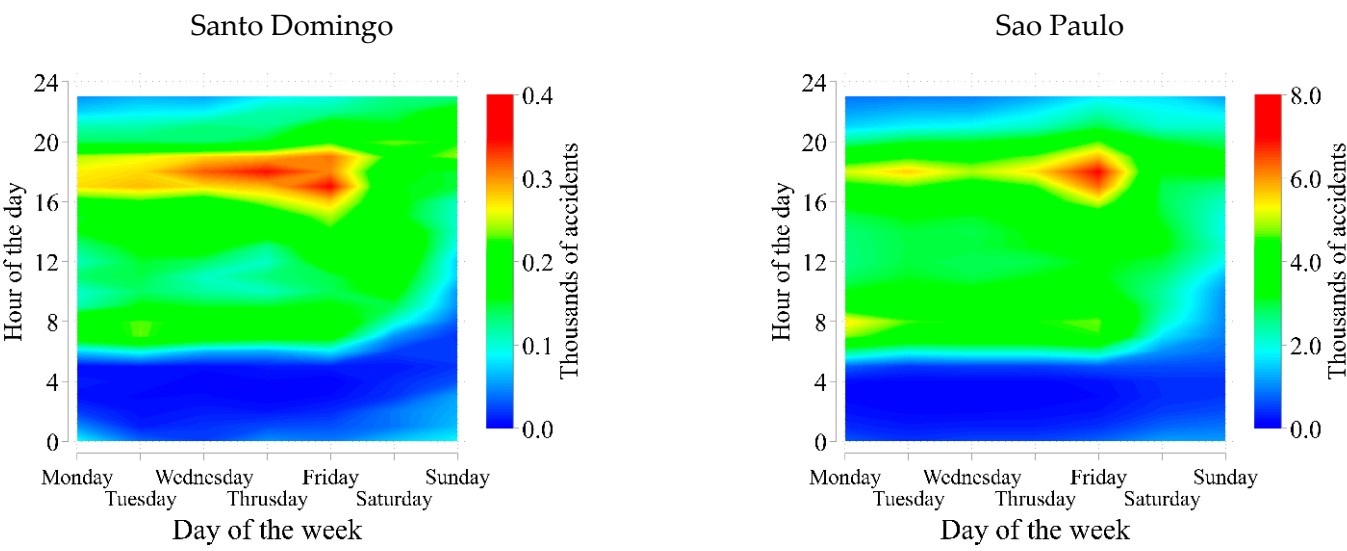

**Figure 4.** Temporal distribution of traffic accidents (selected cities, 2019).

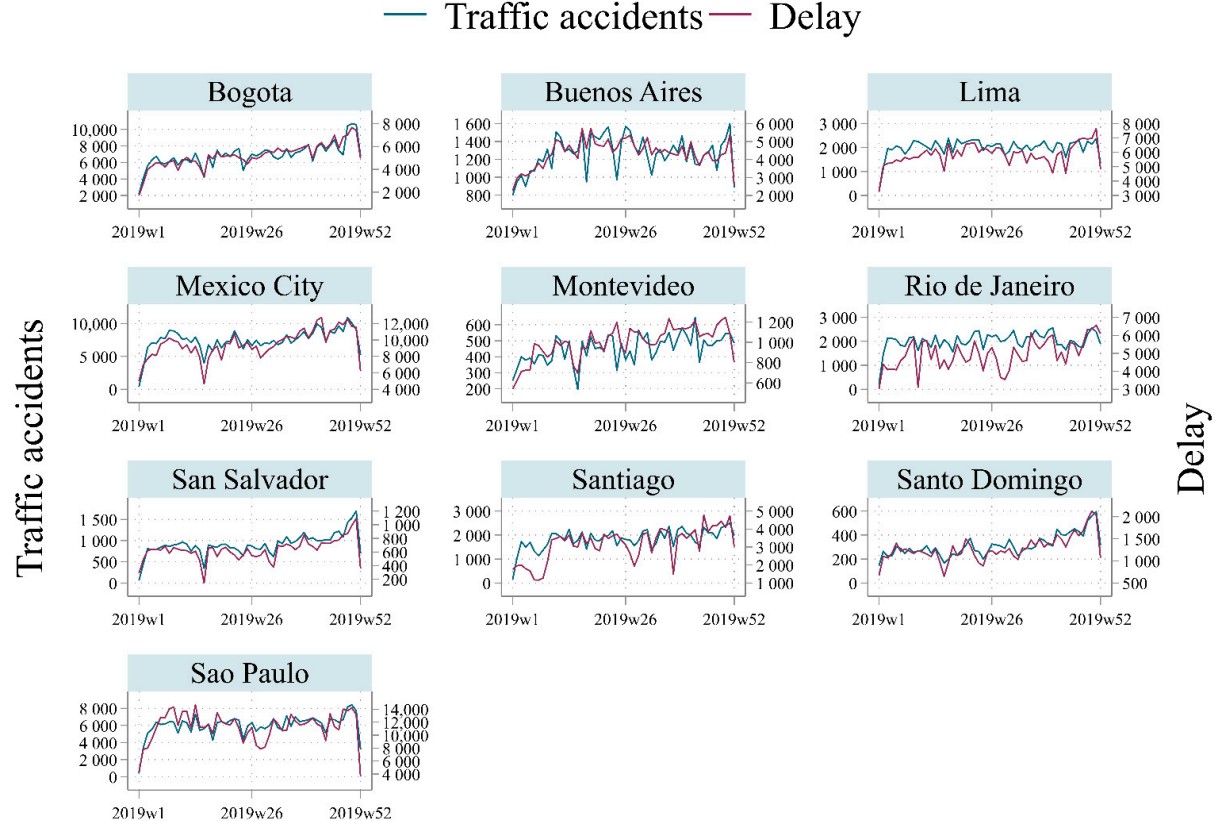

**Figure 5.** Time-series correlation (selected cities, 2019).

The scatter plot in Figure 6 illustrates the correlation between the number of traffic accidents and the level of aggregate delay for each city on an hourly basis. The blue line stands for a linear regression model prediction of traffic accidents controlling by delay and squared delay plus a fixed city effect. Estimations are statistically significant at the 1% level of confidence. Table 2 shows the correlation coefficients with their respective *p*-value according to the Sidak coefficient. Results suggest that a positive relationship exists between

traffic accidents and congestion, with a marginal decrease as delay increases. Bogota and Mexico City are the areas with the highest association, the largest slope, between congestion and number of traffic accidents. In turn, San Salvador, Mexico City, Lima, and Sao Paulo are the areas with the largest correlation (0.86) between congestion and traffic accidents, meaning that both variables covariate symmetrically. While Montevideo presents a small and atypical correlation of 0.21, it is positive and statistically significant, suggesting that although congestion covariates in a smaller scale with traffic accidents, it is still in line with the results obtained for the other cities.

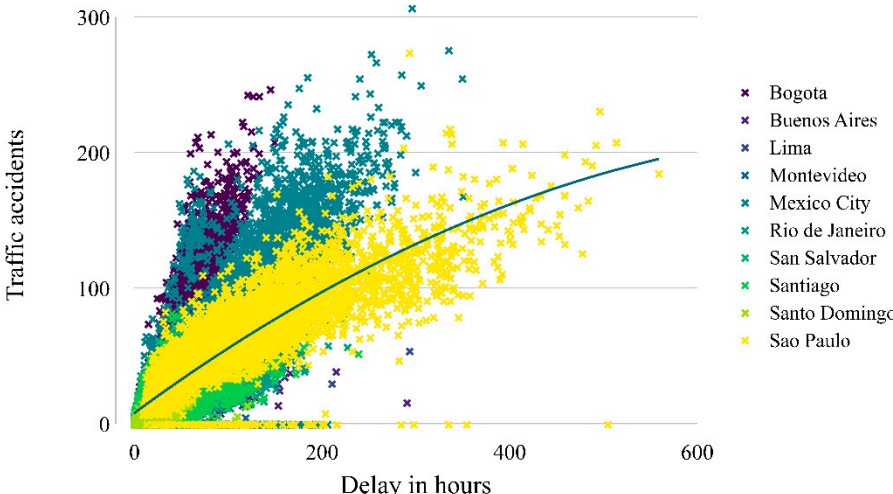

**Figure 6.** Correlation between accidents and delay (selected cities, 2019).

**Table 2.** Correlation coefficients (selected cities, 2019).

| City | Coefficient | *p*-Value |
|---|---|---|
| Bogota | 0.83 | 0.00 |
| Buenos Aires | 0.77 | 0.00 |
| Lima | 0.86 | 0.00 |
| Mexico City | 0.86 | 0.00 |
| Montevideo | 0.21 | 0.00 |
| Rio de Janeiro | 0.79 | 0.00 |
| San Salvador | 0.86 | 0.00 |
| Santiago | 0.72 | 0.00 |
| Santo Domingo | 0.64 | 0.00 |
| Sao Paulo | 0.87 | 0.00 |

Finally, the results of the Poisson model are presented in Table 3, using a robust variance estimator. Result shown are for the final estimation, i.e., the second stage of the two-stage instrumental variable estimation process, performed after the system of equations controlling for endogeneity. While the estimation is obtained using unbalanced panel data, the dataset contains information for 99% of all possible timeframes. This resulted in a total of 50,716 observations. The first column in the table shows the estimated coefficients as presented in the methodology. The second column shows the Incidence-Rate Ratio (IRR), which is included to facilitate understanding the results on how congestion influences accidentality rates, according to the coefficients of our model. We calculate the IRR by adding 1000 h of delay to the accidentality rate at a given congestion level. In this context, 1000 additional hours of delay will increase accidentality rates by 0.4%. Finally, the third column indicates the statistical significance of the coefficient. Figure 7 illustrates the marginal effects as percentage point of change on the quantity of accidents after a reduction of 10% in the congestion level, considering congestion conditions in 2019 and

the control variables mentioned in Section 2. Confidence intervals are included for each city. The aggregate effect is −3.4% and it is statistically significant at the 1% level. When considering cities individually, they also show statistically significant effects of congestion on the number of accidents.

**Table 3.** Estimation results (second stage).

|  | Coefficient | IRR | *p*-Value |
|---|---|---|---|
| Delay | 0.004 | 1.004 | 0.00 |
| Ln(hazard) | 0.555 | 1.741 | 0.00 |
| Tuesday | 0.022 | 1.022 | 0.11 |
| Wednesday | 0.016 | 1.017 | 0.08 |
| Thursday | 0.012 | 1.012 | 0.39 |
| Friday | 0.086 | 1.089 | 0.00 |
| Saturday | 0.131 | 1.139 | 0.00 |
| Sunday | 0.019 | 1.019 | 0.68 |

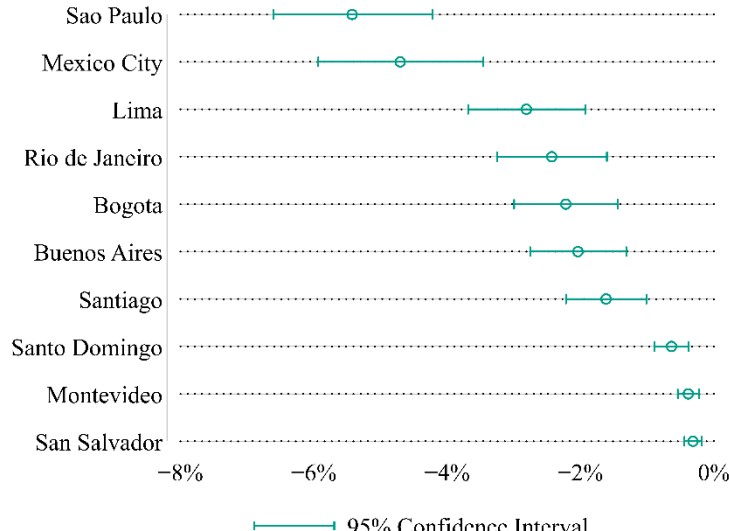

**Figure 7.** Marginal effects (selected cities, 2019).

## 4. Discussion

Our findings suggest that congestion and accidentality are highly correlated. The overall correlation is 0.81, and it is statistically significant. Moreover, congestion has a significant effect on accidentality rates. According to our estimations, 1000 additional hours of delay will increase accidentality rates by 0.4%. This result is not negligible, since a 10% decrease in congestion would reduce traffic accidents by 3.4%. This effect is estimated by taking into account the cities' specific traffic conditions and instrumenting the endogeneity between congestion and accidentality. By doing so, we overcome the twofold challenge faced by extant literature when assessing the effect of urban congestion on accidentality, which we refer to in Section 1. In particular, the granularity of the database allowed us to construct a temporal-spatial partitionable database and apply a Poisson panel data model with an instrumental variable to account for the simultaneous effect of congestion and accidentality in the model.

Results show a positive non-linear effect of traffic delays on road accidents, also after controlling for hazards and roads closed. As such, the effect marginally decreases when delays increase. These findings are related to those of the most recent studies on this subject, which suggest an inversed U-shaped relationship between congestion and accidentality, where the highest level of accidents is reached with an average level of congestion [14].

Analyzing the results obtained for each city in the sample, we find that Sao Paulo and Mexico City would be the cities benefiting the most if congestion decreased by 10%. The number of accidents taking place annually in Sao Paulo and Mexico City would decrease by 5.4% and 4.7%, respectively. Instead, San Salvador, the city with the largest per capita accidentality rate among the ones analyzed in this paper, would obtain a reduction of just 0.3% in the number of accidents. When considering the absolute number of annual accidents, Mexico City would be the area more positively impacted: A 10% reduction in congestion would reduce the number of accidents by 26,000. This reduction would be equivalent to 17 days without accidents in Mexico City. Next in line is Sao Paulo, with a reduction of 23,000, followed by Bogota (11,000); Lima (4000); Rio de Janeiro (3000); Santiago and Buenos Aires (2000); San Salvador (193); Santo Domingo (143); and Montevideo (115).

Our findings are particularly relevant if we take into account the mobility trends in emerging economies and especially in Latin America. The region already holds 4 of the 10 most congested cities in the world (Bogota, Mexico City, Rio de Janeiro, and Sao Paulo) [26]. By 2030, motorization rates are expected to increase by almost 40% [25]. In turn, higher product demand from a growing urban population, together with the boom in e-commerce, is expected to trigger more freight traffic on urban roads [8]. Moreover, while the medium and long-term impacts of Covid-19 are still uncertain, recent studies show short-term changes in modal preferences, increasing private vehicle usage. For example, Basu and Ferreira [27] found that in Boston, 18% of zero-car households intended to purchase a car because of Covid-19, and 26% of them within the following year. In this context, bolder public policies are needed to revert the negative mobility trends, particularly regarding congestion growth [25].

The linkage between congestion and accidentality rates evidenced in this paper can help increase the acceptability of traffic demand policies, particularly when their goal is to reduce car usage. Congestion charging, parking pricing, and low-traffic neighborhoods, among others, are often hard to implement due to the resistance of private vehicle users, residents, and business owners. In these contexts, our findings can aid policymakers in implementing such policies by providing an additional factor that justifies their implementation: Congestion mitigation policies can also help create safer environments.

Likewise, our results help advance research on the relationship between congestion and public health. Wang et al. [8] found that ten additional minutes of commuting time due to congestion is associated with a 0.8% higher chance of suffering from depression. More broadly, the WHO suggested that congestion was related to higher levels of fatigue, alterations to social behavior (more anxiety, for example), communication difficulties, and sleep reduction, ultimately impeding societies' sustainable development [9]. Our study provides additional support to this field of research by evidencing the causal effect of congestion on accidentality. Further research could focus on assessing the relationship between congestion and accidents severity.

Finally, the granularity of the database used in this study allowed the authors to uncover the temporal and spatial dynamics of traffic accidents in the selected cities. These dynamics can provide useful information for the design of traffic management and road safety policies. For example, according to our findings, traffic accidents mostly take place coinciding with traffic peaks in the afternoon, particularly on Fridays. This information can be useful to traffic and police officers, as well as healthcare services, so that they can be better prepared and provide a rapid response to road accidents when and where they are most likely to occur.

## 5. Conclusions

The relationship between congestion and accidentality has been a topic of long-dated interest in academic literature. However, external factors such as weather and road conditions, as well as the bidirectionality of the relationship, have limited scholars' ability to address this topic. Novel technologies in traffic management now provide large amounts

of temporal and spatial disaggregated data on congestion, accidentality, and the different factors influencing the relationship between them. Using over 10 billion observations generated by Waze in 2019 for the ten largest cities in Latin America—Bogota, Buenos Aires, Lima, Montevideo, Santo Domingo, Sao Paulo, Mexico, Rio de Janeiro, San Salvador, and Santiago—and applying both a fixed-effect Poisson regression model for panel data and an instrumental variable approach, we overcome the two main barriers faced by previous research when analyzing the relationship between urban congestion and accidentality. We find a positive non-linear effect of traffic delays on road accidents, with decreasing marginal effects. Our results suggest that a 10% decrease in congestion would reduce traffic accidents by 3.4%. By doing so, more than 72,000 traffic accidents could be avoided annually in the ten cites analyzed here.

While exploring the severity of the accidents was beyond the scope of this research, extant literature shows that reducing congestion would certainly have a positive impact on public health. Further research therefore includes assessing the relationship between congestion and accidents severity. Moreover, most of the ten cities are currently analyzing different transport policies to improve urban mobility and reduce congestion. In this context, further research could also focus on investigating the impact such policies would have on reducing accidentality rates in urban settings. Results from this research could provide policymakers with further support to implement congestion mitigation policies in highly congested urban areas.

**Author Contributions:** Conceptualization, S.S.G., F.B.-M. and A.C.; methodology, S.S.G.; software, S.S.G.; validation, S.S.G. and F.B.-M.; formal analysis S.S.G. and F.B.-M.; investigation, S.S.G.; data curation, S.S.G.; writing—original draft preparation, S.S.G. and F.B.-M.; writing—review and editing, A.C.; visualization, S.S.G.; supervision, A.C.; project administration, A.C. All authors have read and agreed to the published version of the manuscript.

**Funding:** This research received no external funding.

**Institutional Review Board Statement:** Not applicable.

**Informed Consent Statement:** Not applicable.

**Data Availability Statement:** Data acquisition was possible through the program Waze for Cities (https://www.waze.com/es-419/ccp/, accessed on 12 June 2021).

**Conflicts of Interest:** The authors declare no conflict of interest.

## Appendix A

### Bogota

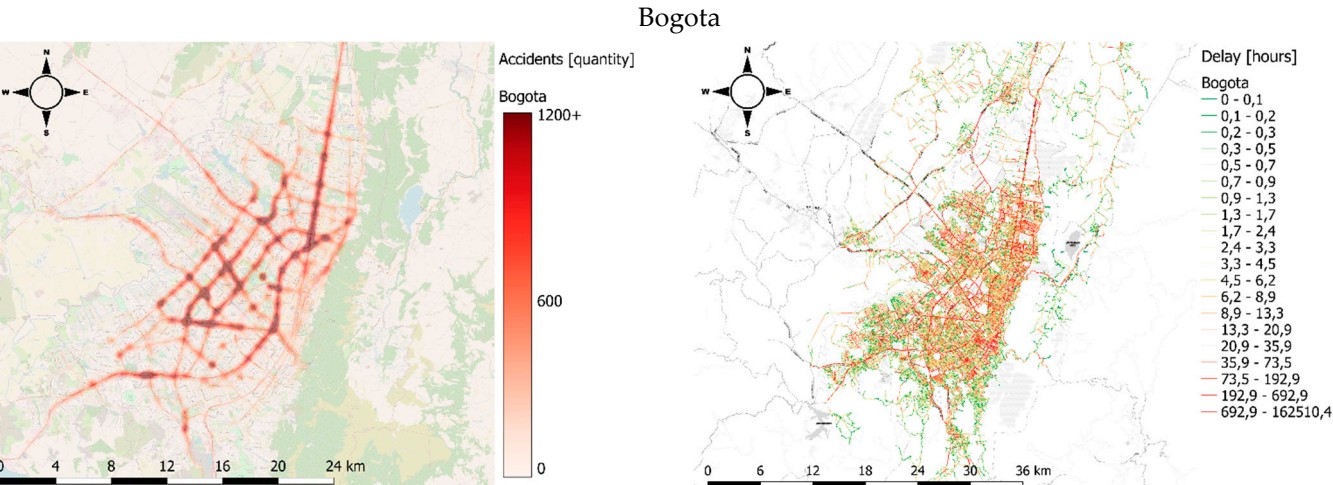

**Figure A1.** *Cont.*

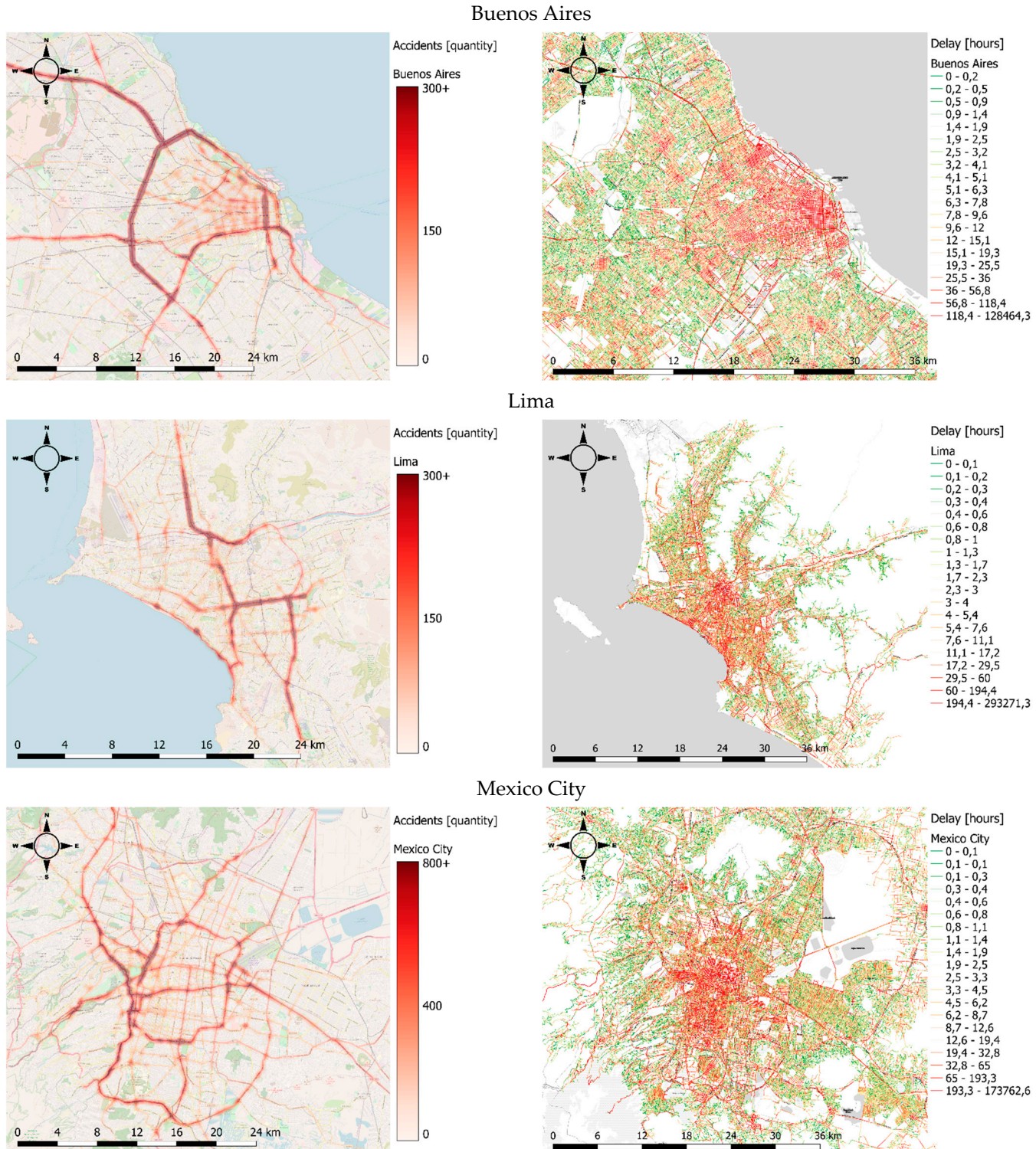

**Figure A1.** *Cont.*

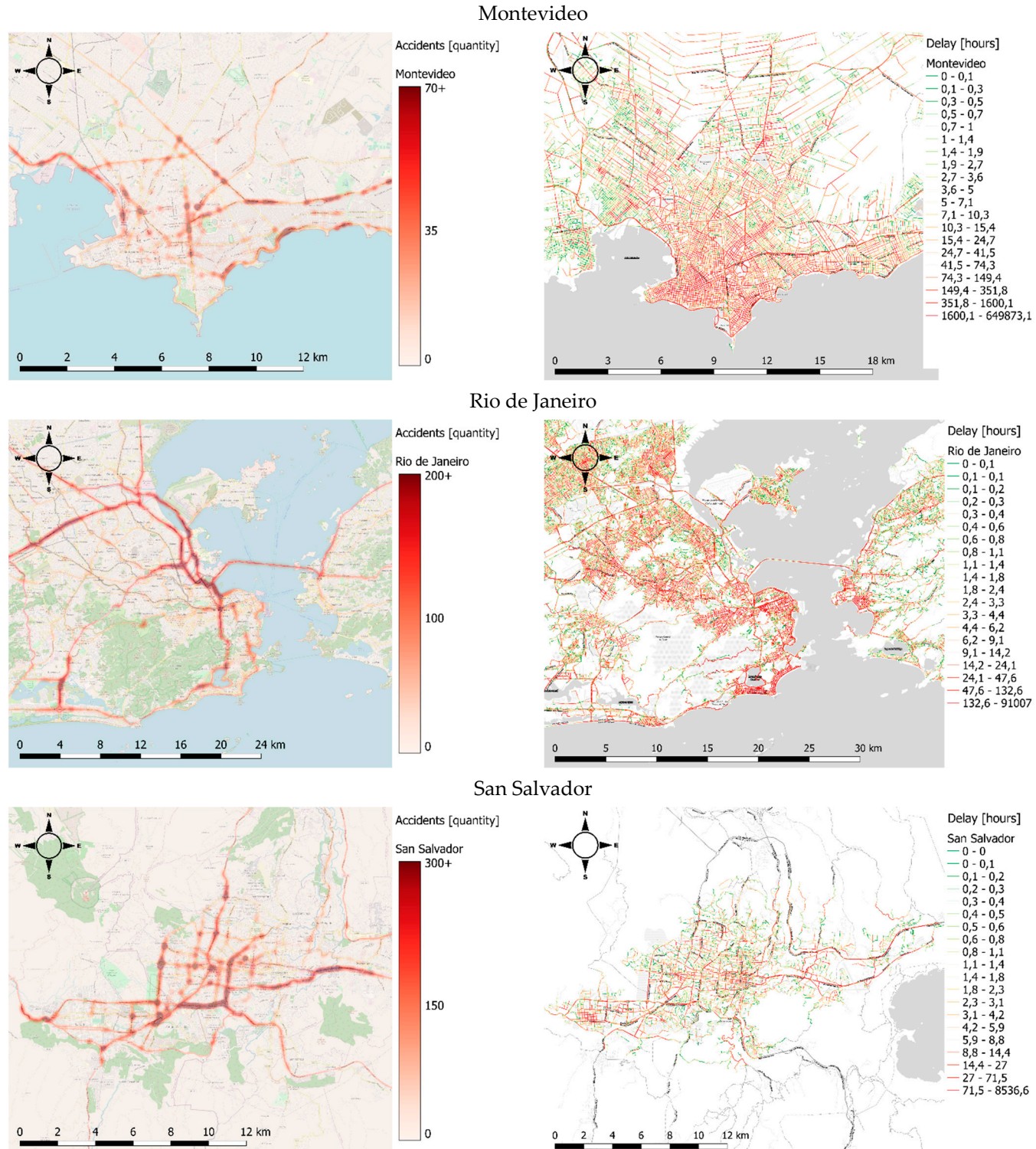

**Figure A1.** *Cont.*

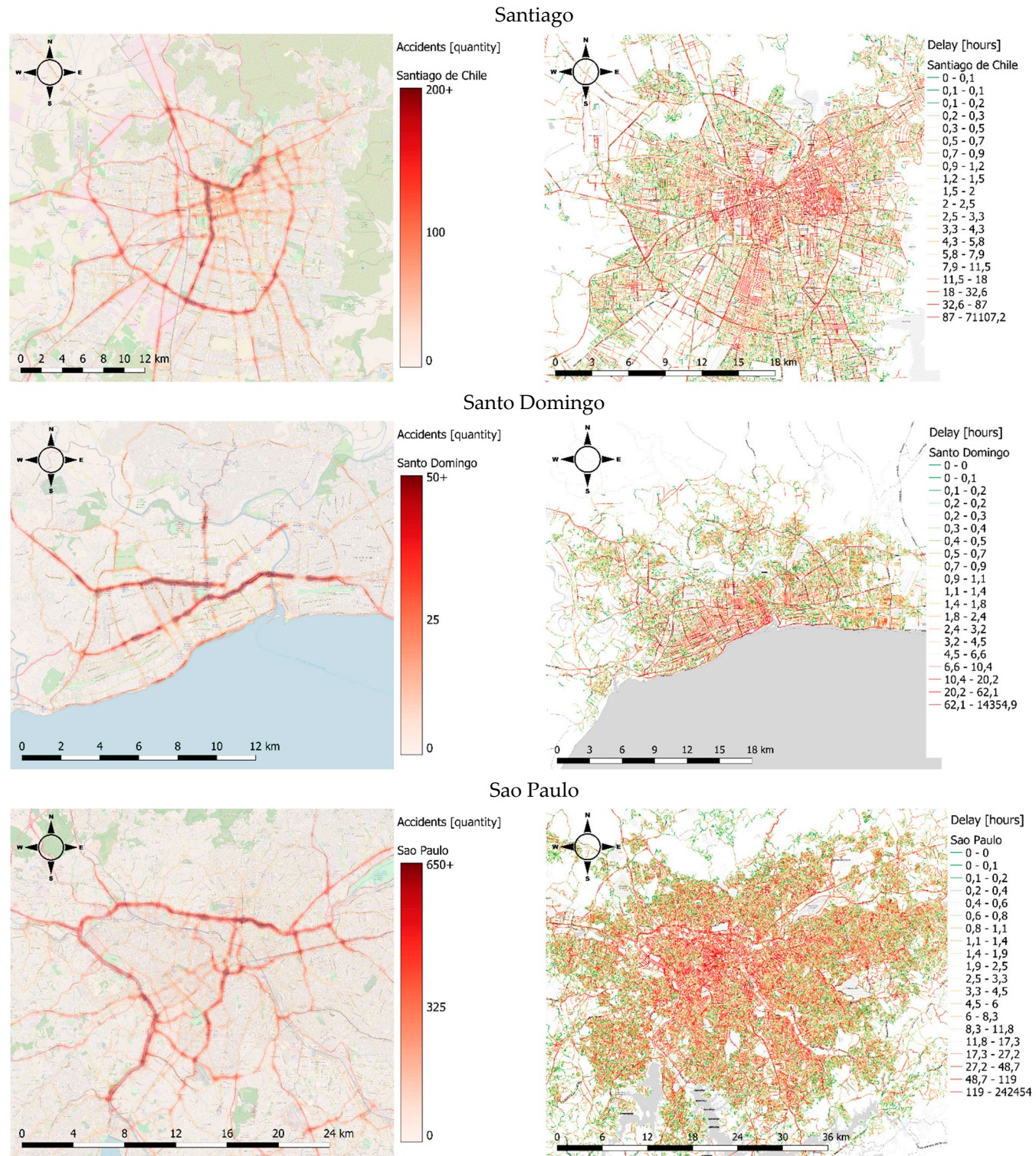

**Figure A1.** Spatial distribution of traffic accidents and delay (selected cities, 2019).

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
