# Peer review of "Understanding the Effect of Traffic Congestion on Accidents Using Big Data"

_sustainability, doi:10.3390/su13137500_

Round 1

Reviewer 1 Report

Dear authors, your paper is nice, well structured and interesting reading. I only have some comments regarding the conclusions; I find it too generalized, dealing with other known issues regarding congestion (same statements mentioned also earlier in the paper) and transport policy (eg. accepting tr. policy measures is an issue per se, I find the relevance of this paper and further research dealing with the congestion and severity of accidents as inputs for policy makers; the measures and their acceptance are different field), with less focus on the research and its results. I suggest rewriting it.

Some minor comments:

Check the explanation of variables after eq. 1. In the line 152 it seems gama and not Y. Also check the numbers of equations and their referencing in the text. 

Line 222: from 3pm to 8pm would be afternoon rather than evening I think.

There were almost 2 million accidents in the analysed year. IS the right axes title in the Figure 3 correct?

Citation in the line 305 is of different style.

Best regards, the reviewer

Reviewer 2 Report

Dear Authors,

Thank you for sending your paper to the journal Sustainability. Your paper is on the topic of the journal, and it is interesting to research. I want to propose to you some improvements to the current version of the paper:

  1. Please try to avoid so-called chain citations. For example [13-15]. Each reference deserved at least two to three sentences.
  2. In chapter 2, add the amount of data and explanation that you have used from WAZE in your research.
  3. Add methodology approach. An exemplary block diagram is always a solution.
  4. Check numbers of formulas and connection to the text. In the text, you have formula 8, but there is no such formula.
  5. “… represent the first stage of the two-stage instrumental variable estimation process …” not clear. Where is the second step?
  6. Do not start the paragraph with: on the table, in the figure, etc.
  7. Overall your figures are excellent and informative, but the current quality is deficient. Please add figures with better quality.
  8. Correlation needs to be explained. For example, type of correlation.
  9. Using of Incidence-Rate Ratio (IRR) needs to be explained.
  10. The discussion part is quite a week, having in mind your research, and please enrich.
  11. In conclusion, you can not make fave: citation: figures, tables, etc. Keep in mind the message that you want to send from the conclusion. The conclusion needs to be rewritten in a manner that you are sending the scientific message: (1) our idea, (2) our methodology, (3) our results, and (4) future research steps.

Regards,

Reviewer 3 Report

About the manuscript:

authors modeled the congestion over the road network to the accidents, for this author used datasets from different South American countries. For the modeling and analysis, the authors adopted the Poisson model. After reviewing the manuscript, the following are my comments.

Comments:

  • The authors used Waze datasets in the present work, and it is not clear how the jam conditions are established.
  • At the same time, it is not clear whether the accidents occur in congested traffic or not?
  • Generally, congestion is a time-space phenomenon. With time it propagates. Did the authors account for any duration of congestion over the road network?
  • In recent studies, researchers highlight that accidents occur more in a free flow regime than in a congested state. How did the currently developed Poisson model account for this?
  • The authors stated that Sao Paulo and Mexico City would benefit the most based on the developed models in the present study. Why, for other cities, only marginal improvement is witnessed. Are there any interferences concerning the infrastructure or any network planning for those cities?
  • I feel the figures are too small, mainly figure 1 and 3.

Round 2

Reviewer 2 Report

Dear Authors,

Thank you for the new version of your paper. I propose that the paper goes to the next step of publishing.

Regards,